# Effect of Bilateral Quadratus Lumborum Block Type I on Patient Satisfaction and Incidence of Chronic Postsurgical Pain Following Cesarean Section-A Randomized Controlled Trial

**DOI:** 10.3390/ijerph18179138

**Published:** 2021-08-30

**Authors:** Marcin Mieszkowski, Marek Janiak, Michał Borys, Paweł Radkowski, Marta Mieszkowska, Yauheni Zhalniarovich, Dariusz Onichimowski

**Affiliations:** 1Department of Anesthesiology and Intensive Care, School of Medicine, Collegium Medicum, University of Warmia and Mazury, Warszawska 30, 10-082 Olsztyn, Poland; pawelradkowski@yahoo.de (P.R.); onichimowskid@wp.pl (D.O.); 2Anesthesiology and Intensive Care Clinical Ward, Voivodal Specialistic Hospital, Żołnierska 18, 10-561 Olsztyn, Poland; 31st Department of Anesthesiology and Intensive Care, Medical University of Warsaw, Lindleya 4, 02-005 Warsaw, Poland; mjaniak8@gmail.com; 42nd Department of Anesthesiology and Intensive Care, Medical University of Lublin, 20-081 Lublin, Poland; michalborys1@gmail.com; 5Department of Surgery and Radiology with Clinic, Faculty of Veterinary Medicine, University of Warmia and Mazury, Oczapowskiego 14, 10-719 Olsztyn, Poland; marta.jaskoolska@gmail.com (M.M.); eugeniusz.zolnierowicz@uwm.edu.pl (Y.Z.)

**Keywords:** quadratus lumborum block type I, cesarean section, ropivacaine, satisfaction, chronic postsurgical pain, multimodal analgesia

## Abstract

Background: Quadratus lumborum block (QLB) provides a reduction in pain scores and opioid consumption after cesarean section (CS). Intrathecal morphine (ITM) is still considered as the gold standard of acute postoperative pain therapy, but it does have some significant side effects. The aim of this clinical study was to evaluate whether performing the quadratus lumborum block type I in patients undergoing CS would be associated with an increased satisfaction of pain therapy and a decreased incidence of chronic postsurgical pain (CPSP). Methods: Sixty patients scheduled for elective CS were enrolled. All patients received spinal anesthesia and were randomly allocated to either the QLB group (received bilateral quadratus lumborum block type I with the use of 24 m mL 0.375% ropivacaine) or the control group (received no block). The level of satisfaction was evaluated using a three-step scale and the answers provided in a questionnaire regarding the patients’ satisfaction with the method of postoperative pain treatment in the first 48 h. After a 6-month period, all patients were interviewed to evaluate the incidence and possible severity of CPSP. Results: Satisfaction scores were significantly lower in the QLB group than in the control group (*p* = 0.0000). There were no significant differences between the QLB and control groups regarding the occurrence of chronic postsurgical pain after 6 months following CS (*p* = 0.102). No statistical differences between the groups were recorded when we compared the results of the questionnaire after a period of 48 h from CS (the number of participants were limited in number). Conclusions: QLB type I is an analgetic option that increased the satisfaction of parturients with pain therapy after CS compared to patients who did not receive the block, and there is a tendency for a lower incidence of CPSP.

## 1. Introduction

Over several decades, there was a steady increase in the percentage of completed births by cesarean section globally [1]. Pain following cesarean section is somatic and visceral in nature and its inadequate treatment has both short- and long-term possible consequences, such as reluctance to feed the newborn, impaired early ambulation, reduced willingness for future pregnancies, or symptoms of chronic pain in the abdomen and pelvis [2]. Therefore, the optimal control of pain remains an important aspect of postoperative care in cesarean section [3]. Regional blocks in the anterolateral abdominal wall performed under ultrasound guidance are currently an important pillar in multimodal analgesia after cesarean section [4]. The most commonly performed regional fascial plane blocks are the transversus abdominis plane block (TAPB) and the quadratus lumborum block (QLB) [5]. Multiple randomized controlled trials and meta-analyses concerning the use of these fascial blocks and wound infiltration techniques have shown a reduction in opioid consumption in the first 24–48 h following a cesarean delivery [6]. Other tools used to evaluate the effectiveness of analgesic therapy are postoperative pain scores, such as the Visual Analogue Scale (VAS) or Numerical Rating Score (NRS), which assume that a result of up to 3 points in an 11-point scale suggests analgesia is appropriate [7]. However, a good NRS or VAS result does not prompt health care providers to reflect on whether the therapy is satisfactory and could be better managed in the future. To date, few authors of randomized, controlled studies in the field of regional anesthesia for cesarean section evaluated patient satisfaction or conducted a survey of the effectiveness of their actions as seen through the eyes of the patients [8]. Moreover, few authors attempted to assess the incidence and severity of chronic postsurgical pain (CPSP) in this group of patients [9].

We hypothesized that QLB type I, as part of a multimodal analgesic treatment, would result in higher patient satisfaction and would decrease the incidence of chronic postsurgical pain, as assessed by the questionnaires taken after 48 h following a cesarean section and at 6 months from surgery in comparison to patients who did not receive a block. The aim of this study was to test these hypotheses and to observe any additional positive effects in postoperative pain treatment in patients undergoing elective cesarean section under spinal anesthesia.

## 2. Materials and Methods

### 2.1. Study Design

The study was approved by the Bioethics Committee at the University of Warmia and Mazury, Poland (reference number 21/2014), and was conducted between September 2014 and September 2015. Sixty patients, with the American Society of Anesthesiologists’ (ASA) physical status scores of I–II, were included. All patients were scheduled for elective cesarean section via a Pfannensteil incision under spinal anesthesia. Written informed consent for their inclusion in the trial was obtained from the patients. The exclusion criteria were: known drug allergies to those used in the trial, infection or erythema at the injection site, pregnancy-induced hypertension, anatomic anomalies affecting the fascial block, coagulation disorders, and a history of paracetamol or opioid abuse. Patients were randomized using a website application (http://www.randomization.com, accessed on 19 August 2014) and a computer-generated table of unallocated numbers, thus determining who would receive a bilateral quadratus lumborum block type I (QLB I group, *n* = 30) or would not receive a block (control group, *n* = 30). On arrival to the operating room, all parturients were started on standard noninvasive blood pressure monitoring, electrocardiography, and pulse oximetry. An intravenous cannula was placed in the hand or forearm. All patients received a standardized spinal anesthesia in the sitting position at the L3–L4 interspace with 12.5 mg of 0.5% hyperbaric bupivacaine (Marcaine Heavy Spinal, AstraZeneca, Wilmington, DE, USA) and 20 mcg of fentanyl (Fentanyl WZF, Polfa Warszawa, Warszawa, Poland). Following this, patients were placed in a supine position with a 15° left uterine displacement position and supplemental oxygen was provided through a facemask at 6 L/min. The cesarean section was allowed to proceed after a T6 sensory blockade to ensure the loss of cold and touch was confirmed. Intravenous crystalloids and ephedrine were administered as needed to treat hypotension. Patients received an intravenous infusion of 10 of IU oxytocin (Gedeon Richter Plc., Budapest, Hungary) after delivery. A dose of 10 mg of metoclopramide was given intraoperatively for prophylaxis of nausea and vomiting. At the end of the cesarean section, patients in both groups received intravenous paracetamol (1 g) (Perfalgan, Bristol Myers Squibb, Bristol, New Zealand).

### 2.2. Interventions

Following the completion of surgery, under standard monitoring, parturients were placed in lateral positions and the injection site was prepared with a povidone iodine solution. Patients who were allocated to the QLB I group received an ultrasound-guided QLB type I block through the lateral approach using the technique first described by Blanco et al. [10]. A 6 MHz convex transducer (BK Flex Focus 400, Peabody, MA, USA) covered with a sterile sheath was placed transversely above the lateral edge of the rectus muscle and then moved to the midaxillary line until three abdominal wall muscles were seen. The internal oblique and transversus abdominis muscles were then followed posterolaterally until the quadratus lumborum muscle was visualized with its attachment to the lateral edge of the transverse process of the fourth lumbar vertebra, and an intermediate layer of the thoracolumbar fascia was identified as a bright hyperechogenic line. A 20 Gauge 100 mm echogenic needle (Stimuplex Ultra 360, B. Braun Melsungen, Germany) was then advanced in plane to the transducer with real-time ultrasound visualization from a medial-to-lateral direction until the point of injection at the lateral border of the quadratus lumborum muscle. A dose of 5 ml of 0.9% saline was injected to verify the correct needle tip position. A further dose of 24 ml of 0.375% ropivacaine (Ropimol, Molteni) per side was injected with repeated negative aspiration at 4 ml aliquots. The procedure was repeated on the opposite side of the abdominal wall.

### 2.3. Pain Management

In the postoperative period, all patients were admitted to the postanesthesia care unit (PACU) and connected to routine monitoring. Nurses providing postoperative care were blinded to the study group allocation. According to the study protocol, all patients received intravenous paracetamol (1 g) at regular 6 h intervals and 5 mg of morphine was administered subcutaneously to patients with a pain intensity of more than 3 in the NRS scale or on demand at 4 h intervals for the next 48 h.

### 2.4. Outcomes

The level of satisfaction with the analgesic treatment was evaluated by a blinded investigator at 6, 12, 18, 24, 36 and 48 h postoperatively using a 3-step scale; 1—highly satisfied, 2—satisfied, and 3—unsatisfied with the analgesia. After 48 h from the completion of surgery, all patients received a questionnaire regarding their satisfaction with the method of postoperative pain treatment. Additional comments from patients were allowed. Following a 6-month period, all patients included in the study were interviewed by phone to evaluate the incidence and possible severity of chronic postsurgical pain. Other relevant outcomes, such as 48 h morphine consumption, postoperative pain severity, time to the first morphine request, the incidence of postoperative nausea and vomiting, and the presence of pruritus, were reported by the authors in a previous study [11].

Our primary outcome measure in this study was the level of patient satisfaction with the treatment of postoperative pain. Our secondary outcome measures included the assessment of chronic postsurgical pain (CPSP) occurrence.

### 2.5. Sample Size and Statistical Analysis

A sample size was calculated based on data from a pilot study previously published and from similar clinical trials [11,12,13]. Assuming a type 1 error of 0.05 and a study power of 0.8, we included 30 patients per group into the study, also allowing for missing data. The collected data were expressed as minimum and maximum values, median (range), mean, and standard deviation. The results were analyzed by using SPSS 19 (SPSS, Chicago, IL, USA) with a level of significance at *p* = 0.05. The Shapiro–Wilk test was used to test for normality. A non-parametric Mann–Whitney U test was used when the obtained values did not follow a normal distribution. An assessment of whether elapsed time had a significant impact on the levels of satisfaction of the studied patients was made and whether similar changes occurred in both the control group and QLB I group. As the assumption of sphericity was not met, a multivariate analysis of variance was made with a Greenhouse–Geisser test. A Chi-square test was used to compare responses collected from the chronic postsurgical pain phone survey. It was not possible to define numerical statistical differences between the groups in the questionnaire given to patients after 48 h of care.

## 3. Results

Sixty patients were recruited and randomly assigned to the QLB I group or control group. However, two patients from the QLB I group were excluded due to postoperative analgesic protocol violations, resulting in 58 patients in the final analysis (Figure 1). The baseline demographic data did not differ between the groups (Table 1).

### 3.1. Satisfaction with Pain Therapy

Evaluation of satisfaction with pain therapy, using the three-step scale (highly satisfied, satisfied, and dissatisfied), showed a statistically meaningful difference between the groups in 6, 12, 18, 24 and 36 h after the cesarean section (*p* < 0.05) However, no difference was seen after 48 h from surgery (*p* = 0.17) (Table 2). Elapsed time from the surgical procedure had a significant impact on the satisfaction of the studied patients in both groups. Moreover, group allocation had a significant influence on the variability of the satisfaction parameter in favor of the QLB group (*p* = 0.000) (Figure 2).

### 3.2. Chronic Postsurgical Pain Assessment

The analysis of the results of the phone survey after 6 months from surgery showed no significant statistical difference in the occurrence of postsurgical pain felt by patients either from the abdominal incision or the viscera (*p* = 0.102) (Table 3).

### 3.3. The Postsurgical Questionnaire after 48 h

The effectiveness of pain therapy was significantly better, as assessed by the patients in the QLB I group when compared with the control group, given the results of the questionnaire after a period of 48 h from surgery (Table 4, Table 5 and Table 6). However, data were not subject to numerical statistical analysis as the number of participants was limited.

## 4. Discussion

Parturients undergoing cesarean section are a unique group of patients in terms of optimizing postoperative pain relief [3]. On the one hand, the surgical technique induces acute somatic and visceral pain with an intensity reported by patients as moderate to severe, and on the other hand, there is a risk associated with the use of analgesic techniques on breastfeeding the newborn, as well as the need for early ambulation [14]. Additionally, there is a concern regarding chronic postsurgical pain with new interest in the effect of regional blocks on its incidence and severity [15].

For decades, both opioid and non-opioid analgetic drugs were the mainstay in the treatment of postoperative pain after cesarean section [16]. However, taking into account a series of clinical studies, as well as meta-analyses, the clinician planning pain treatment should limit the amount of opioid administered in accordance with multimodal strategies [17]. Techniques implementing local anesthetics were used as part of this therapy, such as intravenous infusions, wound infiltration or, above all, ultrasound-guided regional blocks independent of the surgical incision [18]. Currently, as reported by multiple randomized controlled trials and meta-analyses, the QLB block, regardless of the approach and with the possible exception of the intramuscular block, is a very effective tool in combating postoperative pain seen in the reduction in opioid consumption and pain intensity following cesarean section [19]. 

In our study, we investigated whether QLB type I was effective in improving satisfaction with postoperative pain treatment among parturients after cesarean section and its effect on the incidence of postsurgical chronic pain. In the year 2014, when the study was designed, QLB type I with a lateral approach technique was the first described point of injection by Blanco et al. and was, therefore, chosen by the authors [10]. Since then, several new approaches of QLB have been described, but the exact mechanism of the analgesic effect of quadratus lumborum block remains under anatomical, radiological, and neurophysiological investigation [20]. The main focus of most studies assessing the effectiveness of QLB and intrathecal morphine (ITM) is opioid consumption, time to first analgesia, and pain severity in standardized pain scores (NRS, VAS, or NPSI—Neuropathic Pain Symptom Inventory) [1,2,3,4,21]. These studies indicate that most effective method in postoperative pain control after cesarean section remains the administration of ITM also when compared with QLB. Its advantages over ultrasound-guided regional blocks are the ease of use, time needed for administration when performing a spinal block, and the minimum amount of additional equipment and materials used. However, one should remember the possible side effects of intrathecally administered morphine, such as delayed respiratory depression (up to 24 h after administration), pruritus, sedative effect, nausea, and vomiting [22]. Limited numbers of clinical trials evaluate patient satisfaction with the chosen method of postoperative pain control, such as ITM following a cesarean section and its impact on persistent pain [8,23].

Salama et al. assessed satisfaction with analgesia in a three-point scale between patients receiving QLB, patients receiving no block, and a third group of patients receiving intrathecal morphine [8]. In the QLB group, 100% of patients rated highly satisfied and satisfied. These findings are in line with our results which also add up to 100% satisfaction. In comparison, the percentage of patients dissatisfied with the pain treatment without a block in the study by Salama et al. was 16.6% and in our study, it was 13.3%. Despite the statistically significant lower consumption of morphine and NRS scores in the ITM group compared to the control, 30% of patients were dissatisfied with the postoperative pain management. Referring to the high efficacy of intrathecal morphine as seen in multiple published meta-analyses, the last result places the choice of intrathecal morphine under debate [8,24]. The three-point scale for assessing satisfaction with pain treatment was also used by other authors [25,26]. The assessment of postoperative pain therapy (based on the standard, most common criteria) does not necessarily correlate with the feeling of satisfaction among parturients with the proposed treatment. Therefore, according to the authors of this paper, not only the total amount of analgetic drugs used, but also the quality of the method of treating acute postoperative pain should be a priority.

We would like to underline the answers provided in the questionnaire after 48 h of treatment by a subgroup of parturients in our trial. All the patients in the QLB group who had a cesarean section in the past without an ultrasound-guided regional anesthetic block stated that they would like to have QLB performed again for a future cesarean section or other abdominal surgery.

Another rarely assessed, but important aspect of patient quality of life following surgery is the occurrence of chronic postsurgical pain (CPSP), which is considered one of the most common surgery-related complications [23,27]. The definition of CPSP involves postsurgical pain of more than 3-months duration that is new when compared to the preoperative status, and the cause is not secondary to an identifiable reason such as infection [28]. The most important risk factors for developing CPSP in parturients undergoing cesarean section are female gender, young adult age, and the severity and duration of acute postoperative pain [29]. The exact mechanism of CPSP is still under debate and no targeted treatment is available at this time [30]. Although the occurrence of CPSP following cesarean section is lower than for other types of surgery [2], the constant increase in the number of cesarean sections globally makes CPSP a possible growing problem [31]. To date, few studies have been published assessing the possible effect of regional anesthesia on the occurrence of CPSP in patients after cesarean section [2,9,23].

In our study, the incidence of CPSP in the surgical incision area after 6 months from the cesarean section was lower in the QLB I group when compared to the control group (1/28 to 5/30, respectively), but this was not statistically significant (*p* = 0.102). This result correlates with those published by Borys et al., in which statistically significant differences between all groups were not seen after 3 and 6 months from the cesarean section, but such differences were noted only after 1 month [2]. However, authors of the above-mentioned study showed a statistically significant difference in the assessment of chronic pain severity at 1 and 6 months when the QLB group was compared to the control group. We did not assess the intensity of CPSP in this study, which is one of our limitations [2].

The presented study has other limitations. Failure to use patient-controlled analgesia (PCA) methods due to the lack of equipment at the time of the trial and subcutaneous morphine administration might have affected patient satisfaction. Another limitation was the fact that only staff taking care of the parturient after cesarean section (nurses and obstetricians) were blinded, which could have led to bias. A more direct comparison could have been made between the QLB I group and the control group if a sham block was administered. Statistical analysis of the answers to the questionnaire after 48 h from surgery was not performed due to the low number of answers provided by the involved participants.

## 5. Conclusions

The results of this study demonstrate that QLB type I is an analgetic option that increases the satisfaction of parturients with pain therapy following cesarean section when compared to patients that did not receive the block, and there is a tendency for the lower incidence of chronic postoperative pain, although no statistical difference was demonstrated. However, significant differences in opioid consumption or lower pain scores do not necessarily translate into the best quality of postoperative pain treatment, as seen by our patients. Additional tools for satisfaction assessment and appropriate follow up on the incidence and severity of chronic pain should be considered as part of pain therapy efficacy following cesarean sections. Further studies may be needed to focus on other aspects of pain treatment perception.

## Figures and Tables

**Figure 1 ijerph-18-09138-f001:**
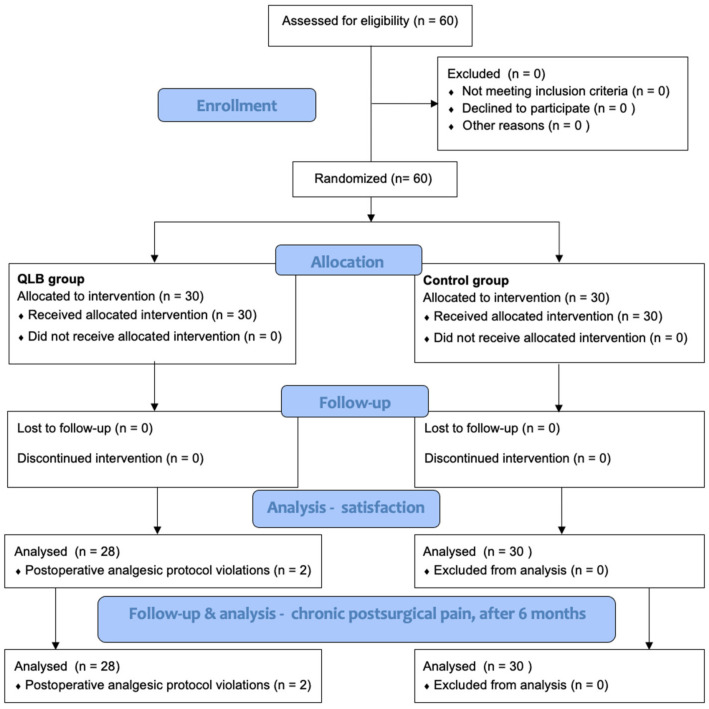
Study flowchart [11].

**Figure 2 ijerph-18-09138-f002:**
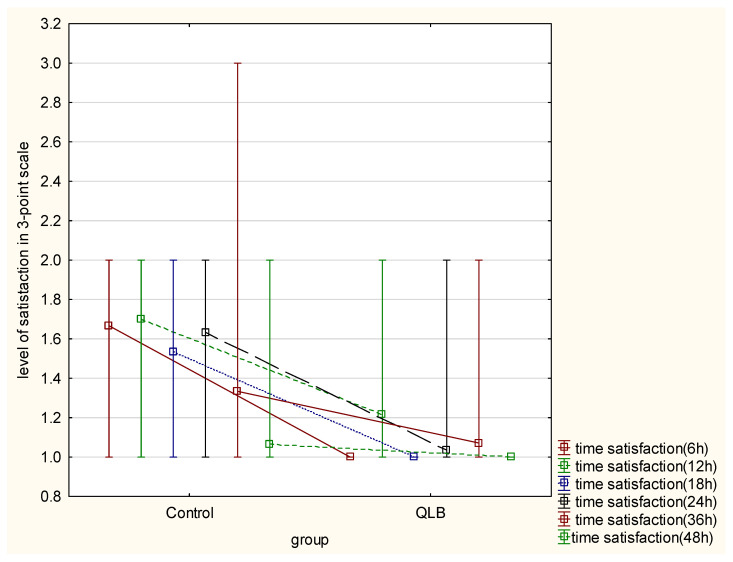
Level of satisfaction with pain therapy in 3-point scale. Presented as figure of interactive mean and interquartile ranges between control group and QLB group, comparing time and level of satisfaction using Greenhouse–Geisser test (*p* = 0.000).

**Table 1 ijerph-18-09138-t001:** Patient demographics. Height, weight, age, and BMI are presented as mean and standard deviations (SD). The groups did not differ in terms of demographic data. Previously published in Mieszkowski et al. Evaluation of the effectiveness of the quadratus lumborum block type I using ropivacaine in postoperative analgesia after a cesarean section—a controlled clinical study (*Ginekol. Pol. 2018*, *89*(2), 89–96) [11].

Group	Control (*n* = 30)		QLB (*n* = 28)		
	Mean	SD	Mean	SD	*p*-Value
Height (cm)	167.80	5.64	166.71	4.93	0.39
Weight (kg)	82.57	14.26	79.96	9.79	0.44
Age (years)	29.29	4.55	28.746	3.25	0.82
BMI (kg/m^2^)	30.63	4.85	30.43	4.09	0.64

**Table 2 ijerph-18-09138-t002:** Evaluation of satisfaction with pain therapy in 3-point scale. The Shapiro–Wilk test was used to test for normality. A non-parametric Mann–Whitney U test was used when the obtained values did not follow normal distribution. Data are shown as medians, range, and mean range.

Group	Control (*n* = 30)			QLB (*n* = 28)			*p*-Value
	Mean Range	Range	Median	Mean Range	Range	Median	
Satisfaction (6 h)	38.83	1165	2	19,50	546	1	<0.05
Satisfaction (12 h)	36.30	1089	2	22,21	622	1	<0.05
Satisfaction (18 h)	36.97	1109	2	21,50	602	1	<0.05
Satisfaction (24 h)	37.87	1136	2	20,54	575	1	<0.05
Satisfaction (36 h)	32.73	982	1	26,04	729	1	<0.05
Satisfaction (48 h)	30.43	913	1	28,50	798	1	0.17

**Table 3 ijerph-18-09138-t003:** The table presents number of patients (No.) who perceived any signs of chronic postsurgical pain at 6 months after cesarean section. A Chi-square test was used to compare responses collected from the chronic postsurgical pain phone survey.

The Occurrence of Chronic Postsurgical Pain 6 Months after CS				
		Yes	No	
	Group			
No.	Control	5	25	30
% of all patients		8.62%	43.10%	51.72%
No.	QLB	1	27	28
% of all patients		1.72%	46.55%	48.28%
*p*-Value		*p* = 0.102		

**Table 4 ijerph-18-09138-t004:** Answer to question no. 1 from the survey. Are you satisfied with the pain management after the cesarean section for the first two days?

Group					
Control			QLB		
* n *	Yes (%)	No (%)	*n*	Yes (%)	No (%)
** 30 **	26 (86.7)	4 (13.3)	28	28 (100)	0

**Table 5 ijerph-18-09138-t005:** Answer to question no. 6 from the survey. Would you like to have a quadratus lumborum block (QLB) performed again in the future for the treatment of postoperative pain?

Group					
Control			QLB		
* n *	Yes (%)	No (%)	*n*	Yes (%)	No (%)
30	Not Applicable		28	28 (100)	0

**Table 6 ijerph-18-09138-t006:** Answer to question no. 8 from the survey. If it was not your first cesarean section in your life and you had a quadratus lumborum block (QLB), do you evaluate pain management better after surgery?

Group					
Control			QLB		
* n *	Yes (%)	No (%)	*n*	Yes (%)	Not Applicable (%)
30	Not Applicable		28	15 (53.6)	13 (46.4)

## Data Availability

The datasets used and/or analyzed during the current study are available from the corresponding author upon reasonable request—Marcin Mieszkowski (marcinm.mieszkowski@gmail.com).

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
