# Peer review of "Effect of Bilateral Quadratus Lumborum Block Type I on Patient Satisfaction and Incidence of Chronic Postsurgical Pain Following Cesarean Section-A Randomized Controlled Trial"

_ijerph, 2021, doi:10.3390/ijerph18179138_

Round 1
Reviewer 1 Report
I appreciate having a chance to review your article. The authors aimed to show the efficacy of QLB1 for patient’s satisfaction regarding postoperative pain management after cesarean section, and the results indicate that QLB1 could improve the satisfaction score. QLB1 is recommended in the recent PROSPECT guideline when intrathecal morphine is not used. The present study can add another benefit of QLB1 regarding patient satisfaction after cesarean section. However, there are some concerns in this manuscript.
Registration
The registration number was missed in the manuscript. Did the author register in a registry before including the patients? If not, this would be the most serious concern for publication. Please clarify this point.
Statistical analysis
Table 2: there are 6 time points and the authors used the Mann-Whitney U test repeatedly. This would increase a familywise error rate. How about applying the Kruskal-Wallis test and Bonferroni correction for the post hoc test? And I could not understand what the mean range and the range meant?
Figure 2: The authors used the Greenhouse-Geisser test for a multivariative analysis for the results of CPSP evaluation. I am not familiar with the analysis and the analysis itself would be right, but why were the IQRs less than 1? This was 3-point scale, so the least number is 1 and it was impossible the IQRs were less than 1.
Did the author have any information about NRS for pain intensity or morphine consumption after surgery? Although the author mentioned the limitation of the present study, I would like to know whether postoperative pain was managed adequately or not. If NRS was higher in the control group, the satisfaction score difference would be derived from the complaint for pain. Or if morphine consumption was similar between the groups, it would not be suggested that QLB1 was performed adequately.
How did the authors evaluate the effect of QLB1? Although I know it was difficult because the influence of spinal anesthesia remained just after the surgery, it would be possible that the improved satisfaction was a result of the placebo effect because the patients could not be blinded. It can be one of the limitations.
How the authors define CPSP? A questionnaire seemed to be used, but I could not know the detail (Only No.1, No.6, and No.8 were shown in the Tables). Did the author diagnose as CPSP when any signs of pain in the questionnaire were found (in Table 3)? If the authors want to show Tables 4-6, you should show all the results for questions no.1-8 and put those 8 results into one table together. Otherwise, your tables seem to be cherry-picking.
Line 242: Tamura, not Salama
Line 369: Ref 19 is from “J Anesth”, not Anesthesia.
Author Response
Dear Reviewer,
We are very pleased that you have undertaken a review of our article. All comments are very valuable and we read them very carefully. Thanks to them, the quality of the article will increase significantly and we hope that we will be able to respond satisfactorily and will be positively received by you.
Responses are numbered with Roman numerals and divided according to the submitted review.
Kind regards,
Authors
I. Registration
The study was approved by the Bioethics Committee at the University of Warmia and Mazury (number 21/2014) before recruiting patients. After obtaining the above-mentioned consent, the study was no longer registered. The main author and responsible for study design of the presented article would like to draw the attention of the Reviewer and the Editorial Board of IJERPH to several important issues. As mentioned in the article, some of the results of the study have already been published (ref no 11) in the Polish Gynecology Journal. From 2018 till date according to the Web of Science research portal, the published article was cited more than 20 times as of 07/08/2021. Moreover, only in 2021 the article was included in the meta-analyzes of the efficacy of QLB in the treatment of acute pain after cesarean section such as: El-Boghdadly et al. [1], Hussain et al. [2] or the latest PROSPECT guidelines by Roofthooft et al. [3]. Considering the above, the authors express their deep hope that this will not be the main obstacle to the publication of the presented article in IJERPH.
- El‐Boghdadly, K.; Desai, N.; Halpern, S.; Blake, L.; Odor, P. M.; Bampoe, S.; Carvalho, B.; Sultan, P.
Quadratus lumborum block vs. transversus abdominis plane block for caesarean delivery: a systematic review and network meta‐Anaesthesia. 2021, 76(3), 393-403. - Hussain, N.; Brull, R.; Weaver, T.; Zhou, M.; Essandoh, M.; Abdallah, F. W.. Postoperative Analgesic Effectiveness of Quadratus Lumborum Block for Cesarean Delivery under Spinal Anesthesia: A Systematic Review and Meta-analysis. Anesthesiology. 2021. 134(1), 72-87.
- Roofthooft, E.; Joshi, G. P.; Rawal, N.; Van de Velde, M.; PROSPECT Working Group* of the European Society of Regional Anaesthesia and Pain Therapy and supported by the Obstetric Anaesthetists’ Association, Joshi, G. P.; Pogatzki-Zahn, E.; Van de Velde, M.; Schung, S.; Kehlet, H.; Bonnet, F.; Rawal, N.; Delbos, A; P. Lavand’homme, H. PROSPECT guideline for elective caesarean section: updated systematic review and procedure‐specific postoperative pain management recommendations. Anaesthesia. 2021, 76(5), 665-680.
II. Statistical analysis
Table 2.
Mean range, range meant are the parameters of the U-Mann-Whitnay statistical test. They appear on printouts of statistical programs. The use of the Kruskal-Wallis test and then the post-hoc test would be possible when comparing and answering the question about the existence of a statistically significant difference between the tested satisfaction levels between individual time points. Therefore, the use of the above test would be correct if we were to test the difference in satisfaction of the examined patients between particular time points. The aim of our study, however, was to study statistically significant differences between the investigated control groups and QLB at individual time points. Therefore, we used a non-parametric test between the two groups.
Figure 2.
Thank you very much for your important remark on Figure 2. The graph has been corrected.
III.
Acute postoperative pain in the conducted study was assessed in patients between groups using several methods, including:
1) morphine consumption with division into time intervals and in total for 24 and 48 hours
2) time from cesarean section to first dose of morphine
3) NRS with division into time intervals and in total for 24 and 48 hours at rest
The results of the obtained analysis of the above assessments were published in the earlier article mentioned by the authors (ref. 11).
Therefore, they could not be re-included in the article under discussion.
The main aim of the authors of the reviewed article was to raise the issue of satisfaction with the applied pain treatment method and the incidence of CPSP. The obtained results indicate that the applied QLB1 blockade improved the satisfaction with the treatment of acute pain after cesarean section in relation to patients without the blockade.
Currently, many meta-analyzes show that spinal morphine remains the most effective in the treatment of acute pain after cesarean section, unless a regional block is used. However, as mentioned in the discussion, the use of ITM recommended as gold standard despite lower NRS values or morphine consumption (objective assessment) compared to patients with QLB blockade did not improve satisfaction with pain treatment and even this satisfaction was worse (subjective assessment). Summing up, it leads to the reflection that despite low NRS values or total morphine consumption, the applied treatment method may be less positively perceived by patients compared to the method where these results are higher but there are fewer side effects than in the case of spinal morphine.
- Mieszkowski, M.; Mayzner-Zawadzka, E.; Tuyakov, B.; Mieszkowska, M.; Żukowski, M.; Waśniewski, T.; Onichimowski, D. Evaluation of the effectiveness of the quadratus lumborum block type I using ropivacaine in postoperative analgesia after a cesarean section—a controlled clinical study. Pol. 2018, 89(2), 89-96
IV.
As mentioned above, in the presented article, the authors undertook the assessment of the effectiveness of QLB1 in the treatment of postoperative pain after cesarean section between the blocking group and the control group by assessing satisfaction on a 3-point scale assessed 6 times within 48 hours of cesarean section and by assessing the incidence of chronic pain.
Additionally, the medical and nursing staff caring for patients after cesarean section was not informed to which group (QLB1 or control) the patient was randomized.
V.
The definition of CPSP taken for this trial involves postsurgical pain of more than 3 months duration that is new when compared to the preoperative status and the cause is not secondary to an identifiable reason like infection [27].
- Schug, S. A.; Lavand'homme, P.; Barke, A.; Korwisi, B.; Rief, W.; Treede, R. D. The IASP classification of chronic pain for ICD-11: chronic postsurgical or posttraumatic pain.Pain. 2019,160(1), 45-52.
The questionnaire with 8 questions, mentioned in the article, was given to the patients to be completed 48 hours after the cesarean section at the ward. It concerned only the issue of acute pain and the assessment of the method of pain treatment used, including satisfaction and possible comments.
On the other hand, information on the presence of CPSP was obtained from the patients through a telephone survey conducted over 6 months after the performance of the cesarean section.
The authors chose from questions 1-8 and placed the answers to questions 1, 6 and 8 mentioned by the reviewer, considering them to be relevant. However, the authors are willing to share all the answers to the questions from the questionnaire, if the editors expressconsent given the scope of an additional amount of words and pages of text.
VI.
Thank you very much for the point. It remains Salam, however, the reference in the text to be cited from number 8, not 24, is changed.
VII.
Thank you very much for pointing out the error in ref 19. Of course it will be corrected.
Reviewer 2 Report
I have read this paper with great interest, and as this further build on a study previously reported in reference 11. I noticed that the flow figure 1 is similar to reference 11 (open source by google scholar, so should be retrievable by the editorial offices also) and the same holds true for the table on clinical characteristics. To avoid copyright and plagiarism, it is important to document that you are allowed to use the same figure and table, or adapt the documents. The editorial offices should further verify this.
related to content, some relevant adaptations are needed:
A recent meta-analysis finds a clinically relevant incidence of CPSP 'wound' after caesarean section ranging from 15% at 3 months to 11% at 12 months or longer that has been largely stable in recent years (Weibel et al, Eur J Anaesthesiol 2016). Statistics: the study has been powered on the short term outcome, with the long term outcome data as safety, secondary parameters. Consequently, the conclusions should be tailored down, as the a priori absence of any differences in a 60 cases cohort, with an overall incidence of 15 %.
Is it correct that this was an open label study for both patient and physician. If so, this could result in bias, and should be further stressed.
Can the authors provide references on the 3 step pain reporting tool used in the study, as this parameter has been analysed subsequently, assuming a continuous, linear relation (0-3 points).
The ‘satification’ has been assessed as a ‘general’ satifaction’, but is there information on pain management itself, versus side effects like itching, or vomiting ?
Author Response
Dear Reviewer,
We are very pleased that you have undertaken a review of our article. All comments are very valuable and we read them very carefully. Thanks to them, the quality of the article will increase significantly and we hope that we will be able to respond satisfactorily and will be positively received by you.
Responses are numbered with Roman numerals and divided according to the submitted review.
Kind regards,
Authors
I.
Study flowchart (Figure 1) differs from that used in the mentioned article (Ref no 11) due to the additional section on "Follow-up & analysis - chronic postsurgical pain, after 6 months". The "study flowchart" style contained in the article is often used by the authors, for example:
- Borys, M.; Zamaro, A.; Horeczy, B.; Gęszka, E.; Janiak, M.; Węgrzyn, P.; Czuczwar, M.; Piwowarczyk, P. Quadratus Lumborum and Transversus Abdominis Plane Blocks and Their Impact on Acute and Chronic Pain in Patients after Cesarean Section: A Randomized Controlled Study. J. Environ. Res. Public Health. 2021, 18, 3500. https://doi.org/10.3390/ ijerph18073500
- Pangthipampai, P.; Dejarkom, S.; Poolsuppasit, S.; Luansritisakul, C.; Tangchittam, S. Bilateral posterior Quadratus Lumborum block for pain relief after cesarean delivery: a randomized controlled trial. BMC Anesthesiol. 2021, 21(1),1-11.
- Yayik, Ahmet Murat, et al. Less painful ESWL with ultrasound-guided quadratus lumborum block: a prospective randomized controlled study. Scandinavian journal of urology, 2019, 53.6: 411-416.
However, if the Editorial Board decides that the layout should be different, the authors will change its style.
We contacted the editors of the Polish Gynecology Journal and as article 11 is open-access, they have no reservations to include patient demographics. The table will be annotated with a footnote identifying the original article.
II.
The incidence of CPSP following cesarean delivery is still unknown. A second meta-analysis including 17 studies has shown its incidence range from 4% to 41.8% at 2 to 6 months (Yimer H, Woldie H. Incidence and associated factors of chronic pain after caesarean section: a systematic review. J Obstet Gynaecol Can. 2019;41(6):840-854). For this reason it is difficult to include this parameter in assessing the power for the study.
III.
The staff taking care of the cases, i.e. nurses and obstetricians, were blinded to minimize bias.
Of course, this may be included in limitations, if this form is deemed appropriate by the reviewer.
IV.
Below, selected publications on the 3 and 5-step pain reporting tool.
5 steps:
Svensson, Ingrid, Björn Sjöström, and Hengo Haljamäe. "Influence of expectations and actual pain experiences on satisfaction with postoperative pain management." European Journal of Pain 5.2 (2001): 125-133.
3 steps:
ÖksÜz, Gözen, et al. Comparison of quadratus lumborum block and caudal block for postoperative analgesia in pediatric patients undergoing inguinal hernia repair and orchiopexy surgeries: a randomized controlled trial. Regional Anesthesia & Pain Medicine, 2020, 45.3: 187-191.
YAYIK, Ahmet Murat, et al. Less painful ESWL with ultrasound-guided quadratus lumborum block: a prospective randomized controlled study. Scandinavian journal of urology, 2019, 53.6: 411-416.
V.
Answering a question about pain treatment versus side effects such data was assessed. No correlation between the satisfaction level and side effect incidence was seen.Round 2
Reviewer 1 Report
I appreciate the authors’ reply.
I have confirmed that registration in a public registry such as ClinicalTrials.gov is not included in the instructions for authors of this journal. I understood the authors’ claim.
However, the authors mentioned that some of the results have already been published. This is another serious concern of publishing the present manuscript because this study is not an original randomized controlled trial, but a secondary use of data from the previous study. The sample size was calculated for another endpoint written in Ref. no. 11, so I cannot believe that the sample size was not appropriate to show the difference the authors would like to show. In addition, the authors used the completely same data in Table 1 shown as ref. no. 11, which should not be allowed because the publisher owns the copyright. I don't think it is recommended to split the data and increase the number of papers. In my opinion, I recommended that the authors should resubmit the present manuscript with those points corrected largely as another manuscript.
Table 2
I could not understand your reply. I guess that the authors showed the sum of ranks as ‘range’. But how about the mean range? If this comma is a colon, I guess that ‘mean range’ means median rank. Anyway, they are not expressed as a range, and I think that these parameters did not make sense.
Evaluation of QLB1
I would like to ask whether the QLB1 correctly induced sensory block on the abdominal wall or not. Although I know it was difficult because the influence of spinal anesthesia remained just after the surgery, it would be possible the QLB itself affected the patients’ satisfaction because the patients could not be blinded. It can be one of the limitations.
The authors replied on how to assess CPSP, but they did not correct the relevant part. This definition should be mentioned in the Method section.
The authors should show all the results for questions no.1-8 and put those 8 results into one table together. Those tables seem to be cherry-picking.
Reviewer 2 Report
no additional comments